# Heterometallic Chain Compounds of Tetrakis(μ-carboxylato)diruthenium and Tetracyanidoaurate

**Masahiro Mikuriya** [1,*], **Yusuke Tanaka** [1], **Daisuke Yoshioka** [1], **Motohiro Tsuboi** [1], **Hidekazu Tanaka** [2] **and Makoto Handa** [2,*]

1 School of Biological and Environmental Sciences, Kwansei Gakuin University, 2-1 Gakuen, Sanda 669-1337, Japan; you06you15@gmail.com (Y.T.); yoshi0431@gmail.com (D.Y.); tsuboimot@kwansei.ac.jp (M.T.)
2 Department of Chemistry, Graduate School of Natural Science and Technology, Shimane University, Matsue 690-8504, Japan; hidekazu@riko.shimane-u.ac.jp
* Correspondence: junpei@kwansei.ac.jp (M.M.); handam@riko.shimane-u.ac.jp (M.H.)

**Abstract:** Heterometallic complexes of tetrakis(μ-carboxylato)diruthenium(II,III) with tetracyanidoaurate(III) [Ru$_2$(RCOO)$_4$Au(CN)$_4$]$_n$ (R = CH$_3$ (**1**), C$_2$H$_5$ (**2**), *i*-C$_3$H$_7$ (**3**), and *t*-C$_4$H$_9$ (**4**)) were synthesized and characterized by C,H,N-elemental analysis and infrared spectroscopy and diffuse reflectance spectroscopy. The molecular structures were determined by a single-crystal X-ray diffraction method. A polymeric arrangement with the Ru$_2$(RCOO)$_4^+$ units alternately linked by Au(CN)$_4^-$ units is formed in these complexes. The *trans*-bridging mode of the Au(CN)$_4^-$ unit for connecting the two Ru$_2$(RCOO)$_4^+$ units was observed for **1** and **4**, while the *cis*-bridging mode of the Au(CN)$_4^-$ unit was observed for **2** and **3**. Magnetic susceptibility data with variable temperature were modeled with a zero-field splitting model ($D$ = 75 cm$^{-1}$) and the presence of weak antiferromagnetic coupling between the Ru$^{II}$Ru$^{III}$ units ($zJ$ = −0.15∼−0.10 cm$^{-1}$) was estimated. N$_2$-adsorption isotherms showed Type II curves with $S_{BET}$ of 0.728—2.91 m$^2$ g$^{-1}$.

**Keywords:** tetrakis(μ-carboxylato)diruthenium; tetracyanidoaurate(III); mixed-valent Ru$^{II}$Ru$^{III}$; crystal structure; magnetic property; heterometallic complex





## 1. Introduction

Dinuclear clusters of ruthenium carboxylates have attracted much attention as spin sources for constructing molecular magnetic materials, because these molecules have $S$ = 1 or 3/2 spins within the dinuclear core depending on the oxidation state of the Ru$_2$ core. Usually, these ruthenium carboxylates exist as ruthenium(II)-ruthenium(II) and ruthenium(II)-ruthenium(III) oxidation states in the dinuclear units. In the case of the Ru$^{II}$-Ru$^{II}$ state, two unpaired electrons exist within the Ru$_2$ core, making the dinuclear complex paramagnetic with a large zero-field splitting parameter, while three unpaired electrons are generated in the accidentally degenerate Ru-Ru bond orbitals in the case of the mixed-valent Ru$^{II}$-Ru$^{III}$ state, resulting as more paramagnetic with a comparatively small $D$ value [1–4]. In the former case, the magnetic moments at room temperature were observed in the range of 2.6–3.2 μ$_B$ per Ru$^{II}$-Ru$^{II}$ unit [2], on the other hand, the observed magnetic moments at room temperature are in the range of 3.6 to 4.4 μ$_B$ per Ru$^{II}$-Ru$^{III}$ unit in the latter case, which is a little higher than that of the spin-only value for the $S$ = 3/2 spins [2]. The axial sites of dinuclear cores are available for many kinds of donor ligands, and coordination polymer formation can be achieved by the introduction of bidentate or multidentate linkers such as $N,N'$-bidentate ligands [3,4]. Among many kinds of donor groups, metal cyanides can be used as unique linkers, having a CN group of a triple bond, which may be expected to have an advantage in communicating electrons between the bridged metal atoms. To date, dicyanidometalate, Ag(CN)$_2^-$ [5] and Au(CN)$_2^-$ [6,7], tricyanidometalate, Cp*Ir(CN)$_3$ [8], tetracyanidometalate, M$^{II}$(CN)$_4^{2-}$ (M$^{II}$ = Ni$^{II}$, Pd$^{II}$, Pt$^{II}$) [9], and

hexacyanidometalate $M^{III}(CN)_6^{3-}$ (M = $Fe^{III}$ [10], $Co^{III}$ [11]) ions have been used for assemblies of rhodium(II) carboxylates. For ruthenium carboxylates, heterometallic assemblies of $Ru^{II}Ru^{III}$ carboxylates with dicyanidoargentate(I) [12], dicyanidoaurate(I) [13], tetracyanidonickelate(II) [14,15], tetracyanidopalladate(II) [16,17], tetracynidoplatinate(II) [18], hexacyanidochromate(III) [19–23], hexacyanidoferrate(III) [19–21,24,25], hexacyanidocobaltate(III) [19–21,24,25] and octacyanidotungstate(V) [26–28] have been reported. In these complexes, the observed magnetic interactions via the cyanidometalate are mostly antiferromagnetic between the 3/2 spins of $Ru^{II}$-$Ru^{III}$ units. In this study, we examined tetracyanidoaurate(III) for the metal assembly of tetrakis(μ-carboxylate)diruthenium(II,III), aiming to develop new molecular magnetic compounds. Tetracyanidometalate has another interesting point of view from coordination chemistry, *trans* and *cis*-orientation of this linker to connect two $Ru_2$ units. In the case of tetracyanidonickelate(II), it was difficult to grow single-crystals for these systems [14,15], although we barely obtained small crystals of the heterometallic compound of $Ru_2(CH_3COO)_4^+$ with $Pt(CN)_4^{2-}$ and X-ray crystallography, using SPring 8 radiation revealed a μ4-bridging of tetracyanidoplatinate(II) to form a layer sheet [18]. In this study, we synthesized and characterized new heterometallic complexes of $Ru^{II}$-$Ru^{III}$ carboxylates $Ru_2(RCOO)_4^+$ (R = $CH_3$, $C_2H_5$, *i*-$C_3H_7$, *t*-$C_4H_9$) with tetracyanidoaurate(III) (Scheme 1). We successfully isolated single crystals of these complexes, and the molecular structures were disclosed by single-crystal X-ray diffraction to reveal the orientation of the linking ligands. Magnetic interactions between the $Ru^{II}$-$Ru^{III}$ units via the linking ligands as well as the adsorption properties for $N_2$ gas were investigated.

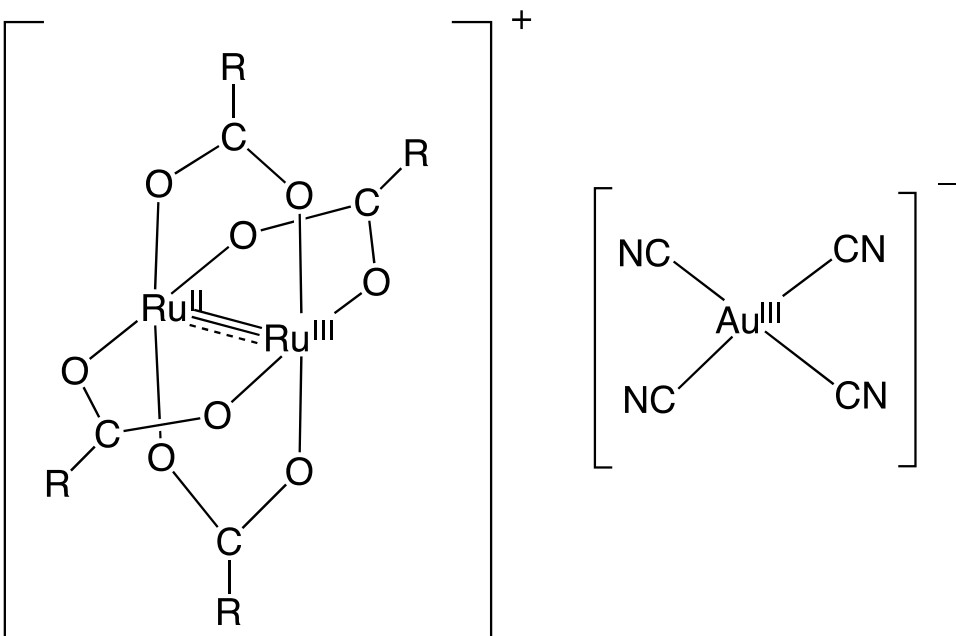

**Scheme 1.** Chemical structures of tetrakis(μ-carboxylato)diruthenium(II,III) cations (R = $CH_3$, $C_2H_5$, *i*-$C_3H_7$, *t*-$C_4H_9$) and tetracyanidoaurate(III) anion.

## 2. Results and Discussion

### 2.1. Synthesis of Heterometallic Compounds of Ruthenium(II,III) Carboxylates with Tetracyanidoaurate(III)

The present complexes were prepared by the reactions of $[Ru_2(RCOO)_4(H_2O)_2]BF_4$ and $K[Au(CN)_4]$ in a 1:1 molar ratio in aqueous solutions as reddish orange precipitate or crystals. The elemental analytical data of the present complexes are in agreement with the 1:1 adduct formulation of $[Ru_2(RCOO)_4Au(CN)_4]_n$.

### 2.2. Infrared Spectra of the Heterometallic Compounds [Ru₂(RCOO)₄Au(CN)₄]ₙ

The IR spectra of the complexes show antisymmetric stretching $\nu_{as}$(COO) and symmetric stretching $\nu_s$(COO) bands at 1437–1487 and 1400–1433 cm$^{-1}$, respectively, with a $\Delta$ value of 32–64 cm$^{-1}$ (Figures S1–S4), which are comparable to those of the starting material [Ru₂(*t*-C₄H₉COO)₄(H₂O)₂]BF₄ ($\nu_{as}$(COO) 1486 cm$^{-1}$ and $\nu_s$(COO) 1425 cm$^{-1}$) with the *syn-syn* mode of μ-carboxylato bridges [29] and in accordance with the structures of tetrakis(μ-carboxylate)diruthenium(II,III) units, as described in Section 2.4. The crystal structure analysis revealed that the Au(CN)₄$^-$ moieties take the *trans*-bridging mode concerning the coordination to the Ru₂(RCOO)₄$^+$ moieties for the acetate complex [Ru₂(CH₃COO)₄Au(CN)₄]ₙ (**1**) and the pivalate complex [Ru₂(*t*-C₄H₉COO)₄Au(CN)₄]ₙ (**4**), while the *cis*-bridging mode was observed for the propionate complex [Ru₂(C₂H₅COO)₄Au(CN)₄]ₙ (**2**) and the isobutyrate complex [Ru₂(*i*-C₃H₇COO)₄Au(CN)₄]ₙ (3). It is recognized that the higher-frequency shift of $\nu$(CN) of metal cyanides is suggestive of bridging CN groups [29]. The CN ion is known to act as a strong σ-donor with a poor π-acceptor. Thus, the σ-donation tends to raise the $\nu$(CN), although the π-backbonding is expected to decrease the $\nu$(CN) [29]. As shown in Figure 1, the CN stretching-vibration bands of the present complexes appeared at 2182–2225 cm$^{-1}$, while the $\nu$(CN) band of K[Au(CN)₄] was observed at 2190 cm$^{-1}$, confirming the bridging of the Au(CN)₄$^-$ moieties to the Ru₂(RCOO)₄$^+$ moieties. The higher-energy $\nu$(CN) (2204 cm$^{-1}$ in **1**, 2225 cm$^{-1}$ in **2**, 2220 cm$^{-1}$ in **3**, 2211 cm$^{-1}$ in **4**) may be ascribed to the bridged CN groups and the lower-energy $\nu$(CN) (2182 cm$^{-1}$ in **1**, 2186 cm$^{-1}$ in **2**, 2186 cm$^{-1}$ in 3, 2182 cm$^{-1}$ in **4**) may be ascribed to the uncoordinated CN groups. It is notable that the energy difference between the higher- and lower-energy CN-stretching bands of the complexes **2** and **3** with the *cis*-bridging mode are considerably larger than those of the complexes **1** and **4** with the *trans*-bridging mode.

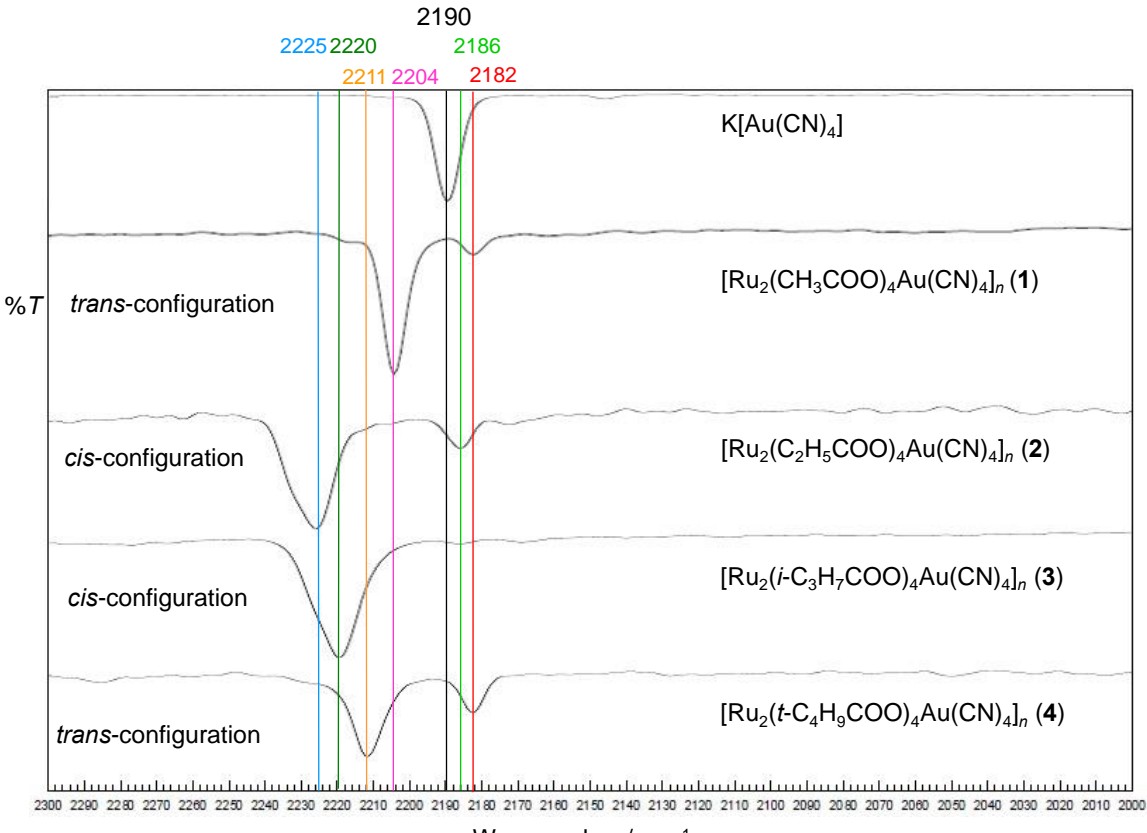

**Figure 1.** Infrared spectra of [Ru₂(CH₃COO)₄Au(CN)₄]ₙ (**1**), [Ru₂(C₂H₅COO)₄Au(CN)₄]ₙ (**2**), [Ru₂(*i*-C₃H₇COO)₄Au(CN)₄]ₙ (**3**), and [Ru₂(*t*-C₄H₉COO)₄Au(CN)₄]ₙ (**4**) in the 2300–2000 cm$^{-1}$ region.

### 2.3. Electronic Spectra of the Heterometallic Compounds [Ru₂(RCOO)₄Au(CN)₄]ₙ

The solid-state diffuse reflectance spectra resemble each other as depicted in Figure 2. The present complexes display a broad band at approximately 220–290 nm with a shoulder at 300–310 nm in the UV region, which may be assigned to charge transfer and d-d bands of the Au(CN)$_4^-$ moieties [30] and LMCT transition of σ(axial ligand)→σ*(Ru₂), respectively [31–33]. In the visible region, a broad absorption appears at 436–452 nm, which may be ascribed to π(Ru-O, Ru₂)→σ*(Ru-O) and π(Ru-O, Ru₂)→π*(Ru₂) transitions [31–33]. A broad absorption at approximately 1000 nm, which can be ascribed to δ(Ru₂)→δ*(Ru₂) transition, and a weak absorption at 1500 nm assignable to π*(Ru₂)→δ*(Ru₂) transition were observed in the NIR region [31–33]. The diffuse reflectance spectra are in accordance with the formation of the heterometallic compounds [Ru₂(RCOO)₄Au(CN)₄]ₙ.

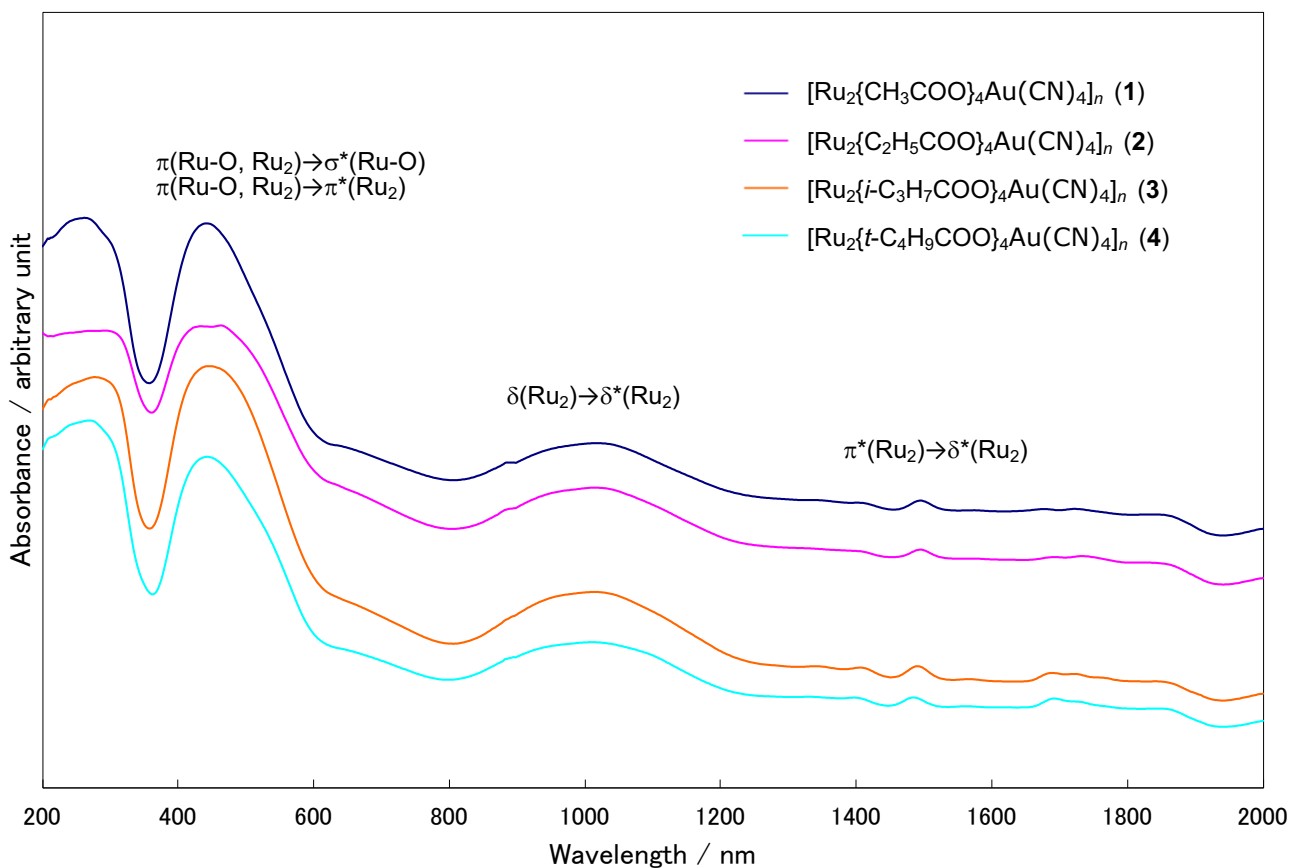

**Figure 2.** Diffused reflectance spectra of [Ru₂(CH₃COO)₄Au(CN)₄]ₙ (**1**) (dark-blue line), [Ru₂(C₂H₅COO)₄Au(CN)₄]ₙ (**2**) (pink line), [Ru₂(*i*-C₃H₇COO)₄Au(CN)₄]ₙ (**3**) (orange line), and [Ru₂(*t*-C₄H₉COO)₄Au(CN)₄]ₙ (**4**) (right-blue line).

### 2.4. Crystal Structures of the Heterometallic Compounds [Ru₂(RCOO)₄Au(CN)₄]ₙ

Single crystals of complexes **1**–**4** suitable for X-ray crystal structure analysis were grown by the slow evaporation of the aqueous solutions of the reaction materials. Crystallographic data are collected in Table 1. Selected bond distances and angles are given in Table S1. The acetate complex [Ru₂(CH₃COO)₄Au(CN)₄]ₙ (**1**) crystallized in the hexagonal lattice. A perspective drawing of the structure of **1** is shown in Figure 3a. The structure consists of a 1-D chain molecule with an alternating arrangement of Ru₂(CH₃COO)$_4^+$ and Au(CN)$_4^-$ moieties, where two cyanido groups of each Au(CN)$_4^-$ moiety are coordinated to the axial sites of two Ru₂(CH₃COO)$_4^+$ moieties in a *trans*-bridging mode. The crystallographic $C_2$ axis contains the N2, C8, Au1, C9, and N3 atoms, forming a square planar Au(CN)$_4^-$ unit with $C_2$ symmetry-related N1 and C7 and C7$^i$ and N1$^i$ atoms, where the superscript i denotes the equivalent position (2−*x*, 1−*x*+*y*, 5/3−*z*). The

Au-C distances are 1.991(4)–2.008(4) Å. Another crystallographic $C_2$ axis contains C6, C5, C1, and C2 atoms, and thus the paddlewheel-type $Ru_2$ core with four *syn-syn* acetato-bridges has the crystallographic $C_2$ axis through the midpoints of Ru1 and Ru1[ii], O1 and O1[ii], and O4 and O4[ii], where the superscript ii denotes the equivalent position ($x$, $x-y$, $7/6-z$). The Ru1-Ru1[ii], Ru1-O(equatorial), and Ru1-N1(axial) distances are 2.2695(7) Å, 2.010(3)—2.023(3) Å and 2.286(3) Å, respectively, which are in the usual range as observed in tetrakis(μ-carboxylato)diruthenium(II,III) clusters [1–4]. The Ru1[ii]-Ru1-N1 angle is 172.34(9)°, deviating from the linear arrangement and causing a wave-like chain structure. This structure is similar to those found in $(PPh_4)_n[Rh_2(RCOO)_4Ag(CN)_2]_n$ (R = $CH_3$, $C_6H_5$, and $C_2H_5OCH_2$) [5] and $(PPh_4)_n[Rh_2(RCOO)_4Au(CN)_2]_n$ (R = $CH_3$, $CH_3OCH_2$, and $C_2H_5OCH_2$) [6]. The crystal structure of **1** is shown in Figure 4a. There are no voids in the crystal structure. The pivalate complex $[Ru_2(t\text{-}C_4H_9COO)_4Au(CN)_4]_n$ (**4**) crystallized in the triclinic lattice. The molecular structure of **4** is similar to that of **1**, taking a *trans*-bridging mode of the tetracyanidoaurate(III) moiety for linking the two tetrakis(μ–pivalato)diruthenium(II,III) moieties, as shown in Figure 3d. The crystallographic inversion centers are located at the midpoints of the Ru1-Ru1[i] and Ru2-Ru2[ii] bonds, where the superscripts i and ii denote the equivalent positions ($1-x$, $-y$, $1-z$) and ($2-x$, $2-y$, $-z$), respectively. The Au-C, Ru-Ru, Ru-$O_{eq}$, and Ru-$N_{ax}$ distances are 1.992(3)–2.005(3) Å, 2.2673(4)–2.2689(5) Å, 2.0132(10)–2.0261(19) Å, and 2.278(2)–2.283(3) Å, respectively, which are similar to those of **1**. The Ru-Ru-$N_{ax}$ angles are 172.75(7) and 173.71(6)°, which are also similar to that of **1**, in accordance with the wave-like chain structure. There are very small voids in the crystal (Figure 4d). The propionate complex $[Ru_2(C_2H_5COO)_4Au(CN)_4]_n$ (**2**) and the isobutyrate complex $[Ru_2(i\text{-}C_3H_7COO)_4Au(CN)_4]_n$ (**3**) crystallized in the monoclinic lattice. The ORTEP views of the molecular structures of **2** and **3** are depicted in Figure 3b,c, respectively. The 1-D chain molecule consisting of alternating $Ru_2(C_2H_5COO)_4^+$ or $Ru_2(i\text{-}C_3H_7COO)_4^+$ and $Au(CN)_4^-$ moieties form in the crystal structures. However, each $Au(CN)_4^-$ moiety takes a *cis*-bridging mode to connect the dinuclear ruthenium moieties, in contrast to the cases for the acetate complex **1** and the pivalate complex **4**, resulting in a zig-zag chain molecule in the propionate complex **2** and the isobutyrate complex **3**. The Au-C, Ru-Ru, Ru-$O_{eq}$, and Ru-$N_{ax}$ distances are 1.995(3)–2.005(3) Å, 2.2665(3) Å, 2.009(2)–2.033(2) Å, and 2.263(2)–2.264(2) Å for **2**, and 1.998(2)–2.000(2) Å, 2.2734(3) Å, 2.0219(15)–2.0270(15) Å, and 2.2728(18) Å for **3**, respectively, which are similar to those of **1** and **4**. The Ru-Ru-$N_{ax}$ angles are 174.96(7)—175.67(6)° for **2** and 175.35(5)° for **3**, which are a little larger than those of **1** and **4**. In these complexes, there are almost no voids in the crystal structures (Figure 4b,c). In these zig-zag chain molecules, two *cis*-cyanido groups of tetracyanidoaurate ion are free from coordination, and were found for the first time in heterometallic compounds of dinuclear metal carboxylates with tetracyanidometalate ions. In $[\{Ru_2(CH_3COO)_4\}_2Ni(CN)_4]_n$ [14], $[\{Ru_2(C_2H_5COO)_4\}_2Pd(CN)_4]_n$ [16], and $[\{Ru_2(CH_3COO)_4\}_2Pt(CN)_4]_n$ [18], four cyanido groups of tetracyanidometalate ion are coordinated to ruthenium atoms to form a 2D sheet in the crystals [16]. 5COO0. In the present complexes, each ruthenium atom takes an equivalent oxidation state of 2.5 between the II and III oxidation states as found in the $Ru^{II}$-$Ru^{III}$ carboxylates reported thus far [1–4].

**Table 1.** Crystallographic data and structure refinement of **1**–**4**.

| Complexes | 1 | 2 | 3 | 4 |
|---|---|---|---|---|
| Chemical formula | $C_{12}H_{12}AuN_4O_8Ru_2$ | $C_{16}H_{20}AuN_6O_8Ru_2$ | $C_{20}H_{28}AuN_4O_8Ru_2$ | $C_{24}H_{36}AuN_4O_8Ru_2$ |
| FW | 739.36 | 795.47 | 851.57 | 907.67 |
| Temperature, $T$ (K) | 90 | 90 | 90 | 90 |
| Crystal system | hexagonal | monoclinic | monoclinic | triclinic |
| Space group | $P6_122$ | $P2_1/n$ | $C2/c$ | $P1$ |
| $a$ (Å) | 11.8315 (10) | 9.1142 (7) | 16.6865 (16) | 9.1719 (8) |
| $b$ (Å) | | 16.7312 (14) | 17.9811 (18) | 9.8063 (9) |

**Table 1.** *Cont.*

| Complexes | 1 | 2 | 3 | 4 |
|---|---|---|---|---|
| $c$ (Å) | 23.2375 (19) | 15.8456 (13) | 9.2176 (9) | 19.6140 (17) |
| $\alpha$ (°) | | | | 77.3160 (10) |
| $\beta$ (°) | | 104.6400 (10) | 104.7720 (10) | 80.9730 (10) |
| $\gamma$ (°) | | | | 88.4590 (10) |
| $V$ (Å³) | 2817.1 (5) | 2337.9 (3) | 2674.3 (5) | 1699.7 (3) |
| $Z$ | 6 | 4 | 4 | 2 |
| $D_{calcd}$ (g cm⁻³) | 2.615 | 2.260 | 2.115 | 1.774 |
| Crystal size (mm) | $0.72 \times 0.23 \times 0.15$ | $0.45 \times 0.13 \times 0.11$ | $0.55 \times 0.15 \times 0.14$ | $0.37 \times 0.23 \times 0.13$ |
| $\mu$ (mm⁻¹) | 9.428 | 7.582 | 6.636 | 5.226 |
| $\theta$ range for data collection (°) | 1.988–28.804 | 1.802–28.780 | 1.696–28.759 | 1.077–28.845 |
| Reflections collected/unique | 17,112/2301 | 14,237/5588 | 8386/3190 | 10,479/7764 |
| $[R_1(I > 2\sigma(I)); wR_2$ (all data)] [a] | $R_1 = 0.0153$ $\omega R_2 = 0.0380$ | $R_1 = 0.0207$ $\omega R_2 = 0.0512$ | $R_1 = 0.0156$ $\omega R_2 = 0.0383$ | $R_1 = 0.0230$ $\omega R_2 = 0.0584$ |
| GOF | 1.136 | 1.100 | 1.118 | 1.042 |

(a) $R_1 = \sum ||F_o| - |F_c||/\sum|F_o|$; $wR_2 = [\sum \omega (F_o^2 - F_c^2)^2/\sum (F_o^2)^2]^{1/2}$.

(**a**)

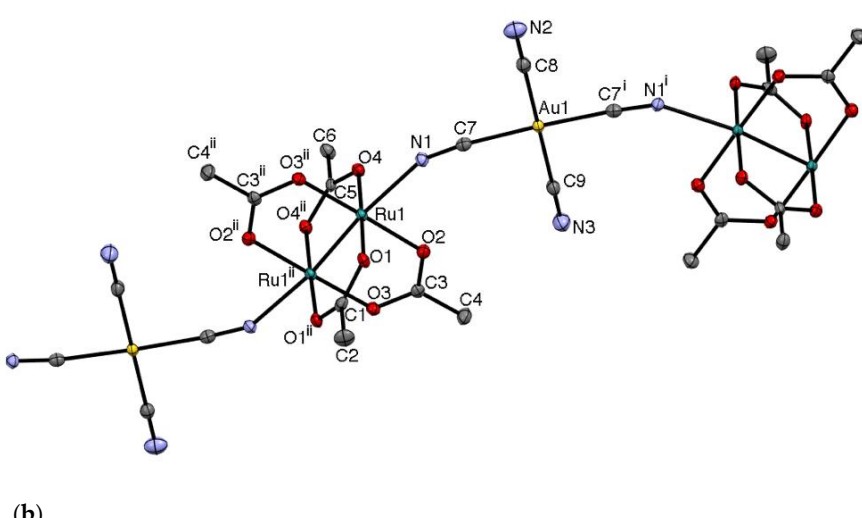

(**b**)

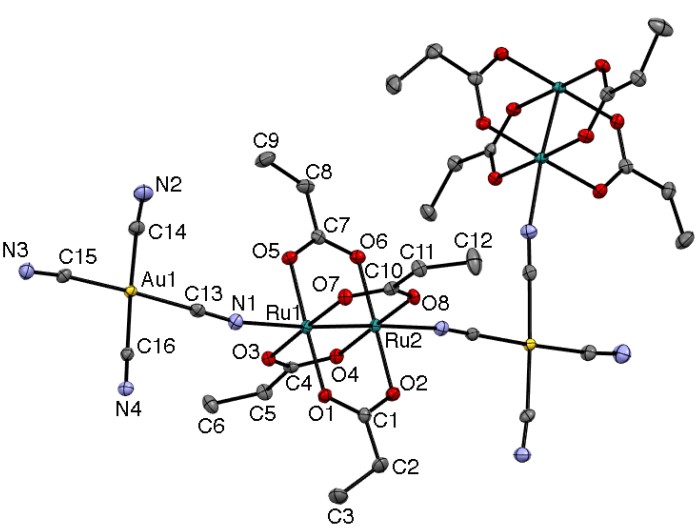

**Figure 3.** *Cont.*

(**c**)

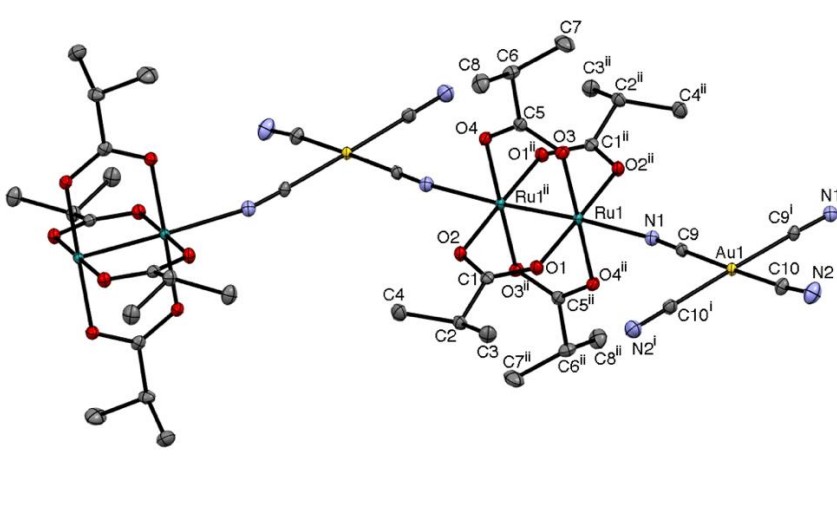

(**d**)

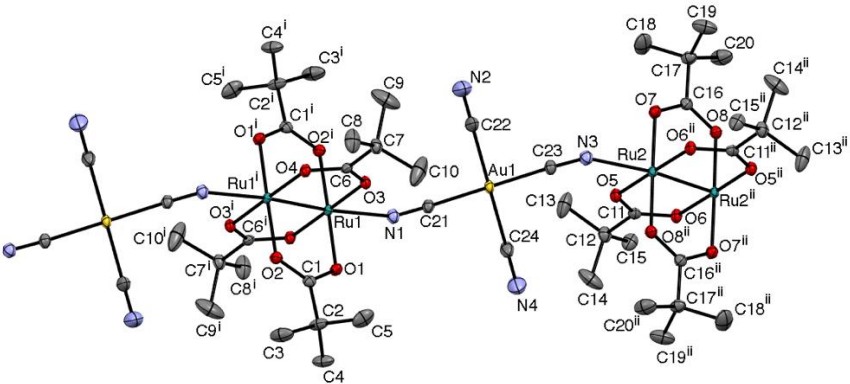

**Figure 3.** The ORTEP view of molecular structures of (**a**) [Ru$_2$(CH$_3$COO)$_4$Au(CN)$_4$]$_n$ (**1**), (**b**) [Ru$_2$(C$_2$H$_5$COO)$_4$Au(CN)$_4$]$_n$ (**2**), (**c**) [Ru$_2$(*i*-C$_3$H$_7$COO)$_4$Au(CN)$_4$]$_n$ (**3**), and (**d**) [Ru$_2$(*t*-C$_4$H$_9$COO)$_4$Au(CN)$_4$]$_n$ (**4**), with thermal ellipsoids (50% probability level). The hydrogen atoms have been omitted for clarity.

(**a**)

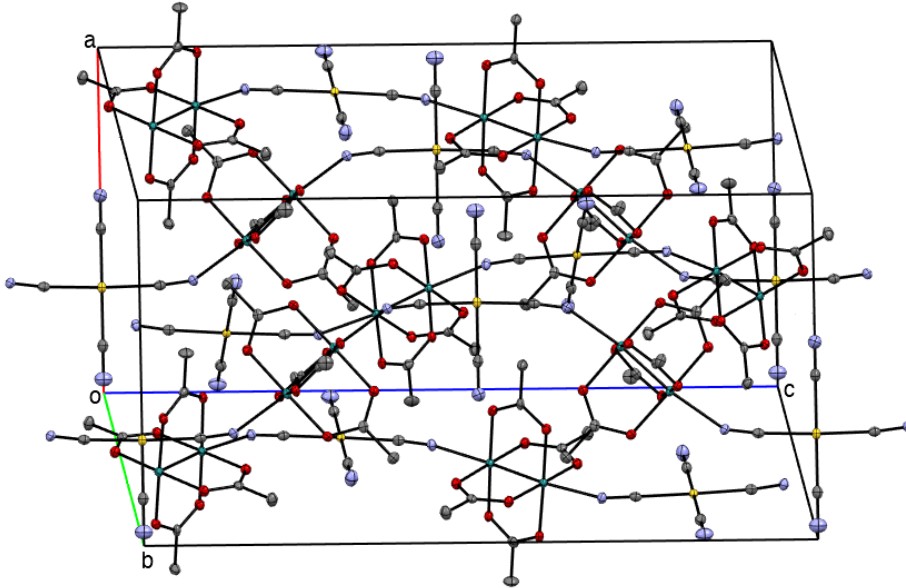

(**b**)

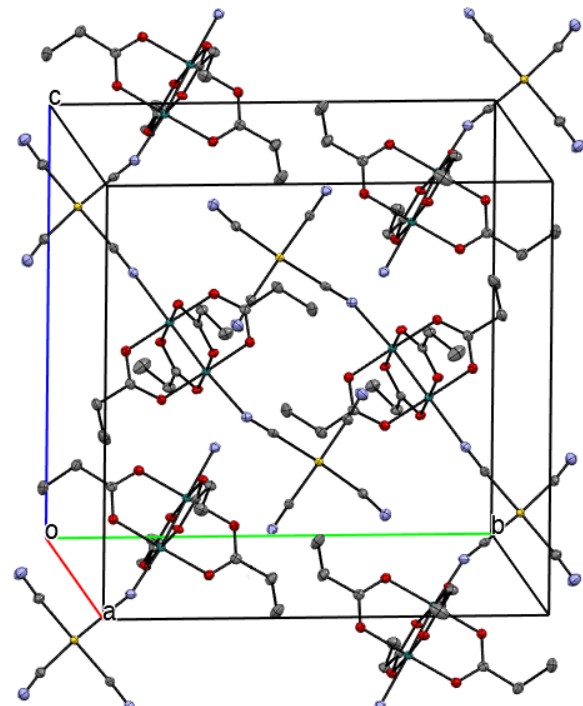

**Figure 4.** *Cont.*

(**c**)

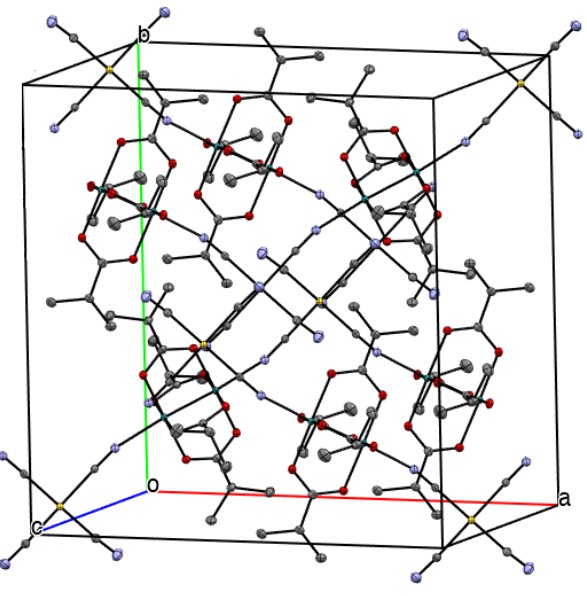

(**d**)

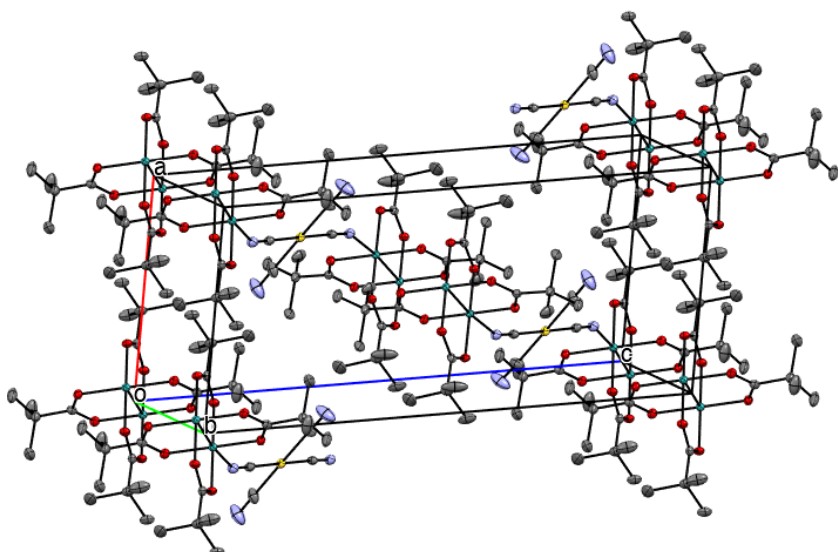

**Figure 4.** Packing diagrams of (**a**) [Ru$_2$(CH$_3$COO)$_4$Au(CN)$_4$]$_n$ (**1**), (**b**) [Ru$_2$(C$_2$H$_5$COO)$_4$Au(CN)$_4$]$_n$ (**2**), (**c**) [Ru$_2$(*i*-C$_3$H$_7$COO)$_4$Au(CN)$_4$]$_n$ (**3**), and (**d**) [Ru$_2$(*t*-C$_4$H$_9$COO)$_4$Au(CN)$_4$]$_n$ (**4**). The hydrogen atoms have been omitted for clarity.

*2.5. Magnetic Properties of the Heterometallic Compounds [Ru$_2$(RCOO)$_4$Au(CN)$_4$]$_n$*

Figure 5 shows the variable-temperature magnetic moment in the measured 4.5–300 K temperature range for [Ru$_2$(CH$_3$COO)$_4$Au(CN)$_4$]$_n$ (**1**) as a representative example. The magnetic properties of the present complexes are similar to each other (Figures S5–S7 for **2**, **3**, and **4**, respectively). The magnetic moments (per Ru$^{II}$-Ru$^{III}$ unit) at 300 K of **1**, **2**, **3**, and **4** are 4.24, 4.26, 4.37, and 4.29 μ$_B$, respectively, which are close to each other, suggesting the presence of three unpaired electrons per Ru$^{II}$-Ru$^{III}$ unit with an *S* = 3/2 state, although the moment values are a little higher the spin-only value, as in most cases for the reported

mixed-valent Ru$^{II}$-Ru$^{III}$ carboxylates, where the magnetic moments were observed in the range of 3.6–4.4 $\mu_B$ at room temperature [1–4]. A decrease in the magnetic moments was observed with decreasing temperature, followed by a further steep decrease close to 5 K, which can be ascribed to the zero-field splitting parameter (*D*) within the Ru$_2$(RCOO)$_4^+$ unit and the antiferromagnetic interaction between the Ru$_2$(RCOO)$_4^+$ units through the axial Au(CN)$_4^-$ linker. The magnetic data were simulated using Equations (1)–(4) described below for the *S* = 3/2 system with the zero-field splitting parameter *D* and the magnetic interaction between the Ru$_2$(RCOO)$_4^+$ units being taken into account by the mean-field approximation [4,34–36]:

$$\chi' = \chi/\{1 - (2zJ/Ng^2\mu_B^2)\chi\} \tag{1}$$

where *z* is the number of interacting neighbors, *J* is the magnitude of the intermolecular interactions, and $\chi$ is the magnetic susceptibility.

$$\chi = (\chi_{//} + 2\chi_\perp)/3 \tag{2}$$

where $\chi_{//}$ and $\chi_\perp$ are magnetic susceptibility terms defined as follows:

$$\chi_{//} = (Ng^2\mu_B^2/kT)\{1 + 9\exp(-2D/kT)\}/4\{1 + \exp(-2D/kT)\} \tag{3}$$

$$\chi_\perp = (Ng^2\mu_B^2/kT)[4 + (3kT/D)\{1 - \exp(-2D/kT)\}]/4\{1 + \exp(-2D/kT)\} \tag{4}$$

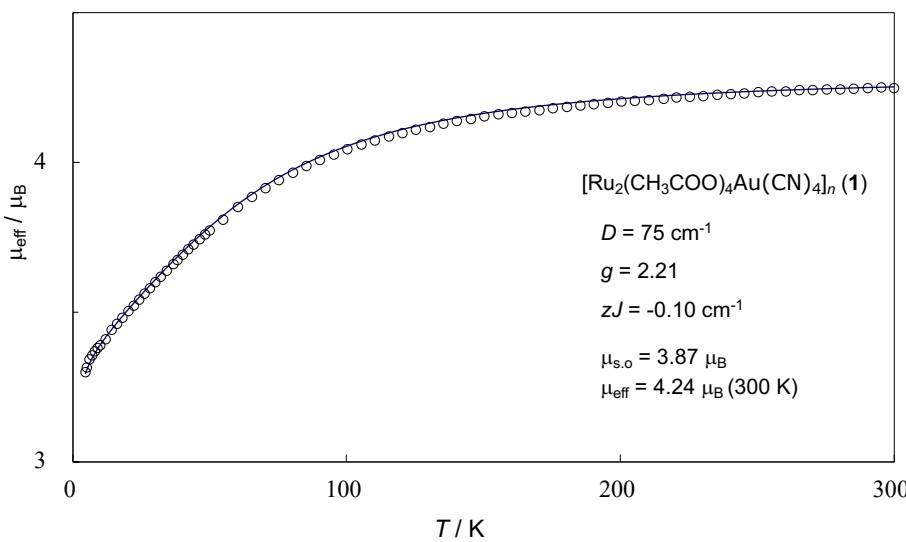

**Figure 5.** Variable temperature of magnetic moment $\mu_{eff}$ for [Ru$_2$(CH$_3$COO)$_4$Au(CN)$_4$]$_n$ (**1**). The solid black line was calculated and drawn with the parameter values described in the text.

The simulation gave the following parameter values: *g* = 2.21, *D* = 75 cm$^{-1}$, *zJ* = −0.10 cm$^{-1}$ for **1**, *g* = 2.21, *D* = 75 cm$^{-1}$, *zJ* = −0.10 cm$^{-1}$ for **2**, *g* = 2.23, *D* = 75 cm$^{-1}$, *zJ* = −0.10 cm$^{-1}$ for **3**, *g* = 2.24, *D* = 75 cm$^{-1}$, *zJ* = −0.15 cm$^{-1}$ for **4**. The obtained *D* values are normal for mixed-valent Ru$^{II}$-Ru$^{III}$ carboxylates and their derivatives [4]. The small *zJ* values mean that the magnetic interaction through the tetracyanidoaurate(III) linker is very weak and it was difficult to differentiate the magnetic interaction through the *cis*- and *trans*-linkers. Similar weak antiferromagnetic interactions were observed in the related heterometallic complexes of ruthenium(II,III) carboxylate with dicyanidoargentate(I) (*zJ* = −0.10, −0.50 cm$^{-1}$) [12], tetracyanidonickelate(II) (*zJ* = −0.20 cm$^{-1}$) [15], tetracynidopalladate(II) (*zJ* = −0.10 cm$^{-1}$) [17], and tetracyanidoplatinate(II) (*zJ* = −0.10 cm$^{-1}$) [18].

### 2.6. N$_2$-Adsorption Properties of the Heterometallic Compounds [Ru$_2$(RCOO)$_4$Au(CN)$_4$]$_n$

The adsorption properties of the present complexes were measured for N$_2$ at 77 K and the isotherms are given in Figure 6 for **1** and Figures S8–S10 for **2**, **3**, and **4**, respectively.

The adsorption isotherms of the present complexes are similar to each other and considered to be of Type II behavior (IUPAC classification) with $S_{BET}$ of 0.728 m$^2$ g$^{-1}$ for **1**, 1.75 m$^2$ g$^{-1}$ for **2**, 2.91 m$^2$ g$^{-1}$ for **3**, and 1.49 m$^2$ g$^{-1}$ for **4**. The nonporous properties are in accordance with the crystal structures of the present complexes.

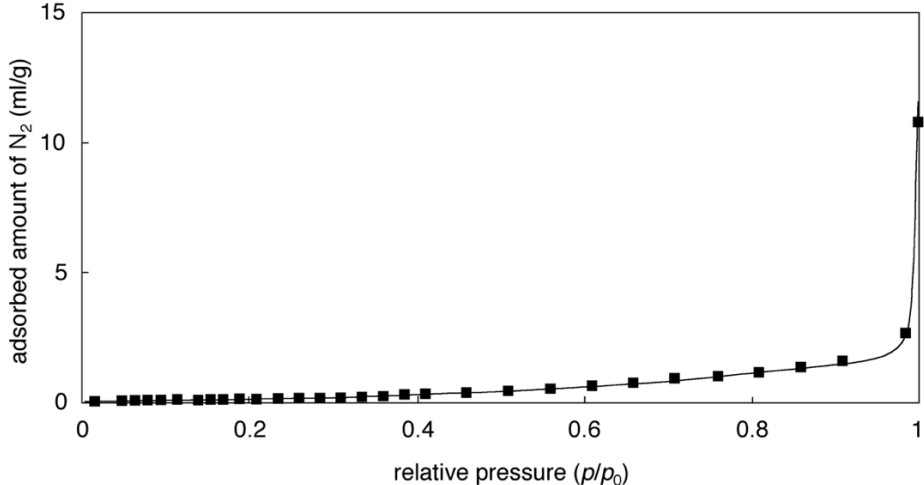

**Figure 6.** Nitrogen adsorption isotherm of [Ru$_2$(CH$_3$COO)$_4$Au(CN)$_4$]$_n$ (**1**). The solid line is a guide for the eye.

## 3. Materials and Methods

All of the reagents and solvents were purchased from commercial sources and used without further purification.

The precursor complexes [Ru$_2$(CH$_3$COO)$_4$(H$_2$O)$_2$]BF$_4$ [12], [Ru$_2$(C$_2$H$_5$COO)$_4$(H$_2$O)$_2$]BF$_4$, [Ru$_2$(*i*-C$_3$H$_7$COO)$_4$(H$_2$O)$_2$]BF$_4$, and [Ru$_2$(*t*-C$_4$H$_9$COO)$_4$(H$_2$O)$_2$]BF$_4$ [28] were prepared by the methods described in the literature.

Synthesis of [Ru$_2$(CH$_3$COO)$_4$Au(CN)$_4$]$_n$ (**1**): To an aqueous solution (5 cm$^3$) of [Ru$_2$(CH$_3$COO)$_4$(H$_2$O)$_2$]BF$_4$ (50.0 mg, 0.0891 mmol), an aqueous solution (5 cm$^3$) of K[Au(CN)$_4$] (30.5 mg, 0.0897 mmol) was added. The solution was stirred overnight. The resulting reddish-brown precipitate was collected, washed with water, and desiccated in vacuo. Yield: 48.7 mg, 72.2%. Found C 19.21, H 1.81, N 7.21%. Calcd for C$_{12}$H$_{14}$AuN$_4$O$_9$Ru$_2$ ([Ru$_2$(CH$_3$COO)$_4$Au(CN)$_4$]·H$_2$O): C 19.03, H 1.86, N 7.40%. IR (KBr, cm$^{-1}$): $\nu$(CN) 2204, 2182; $\nu_{as}$(COO) 1437, $\nu_s$(COO) 1400. Diffuse reflectance spectra: $\lambda_{max}$ 266, 304 sh, 450 br ($\pi$(Ru-O, Ru$_2$)→$\sigma$*(Ru-O); $\pi$(Ru-O, Ru$_2$)→$\pi$*(Ru$_2$)), 1026 ($\delta$(Ru$_2$)→$\delta$*(Ru$_2$)), 1502 ($\pi$*(Ru$_2$)→$\delta$*(Ru$_2$)) nm.

Synthesis of [Ru$_2$(C$_2$H$_5$COO)$_4$Au(CN)$_4$]$_n$ (**2**): This compound was prepared by the reaction of [Ru$_2$(C$_2$H$_5$COO)$_4$(H$_2$O)$_2$]BF$_4$ (10.1 mg, 0.0164 mmol) and K[Au(CN)$_4$] (5.5 mg, 0.016 mmol) in a similar manner to that of **1**. Yield: 4.6 mg, 35%. Found C 24.20, H 2.46, N 7.01%. Calcd for C$_{16}$H$_{20}$AuN$_4$O$_8$Ru$_2$: C 24.16, H 2.53, N 7.04%. IR (KBr, cm$^{-1}$): $\nu$(CN) 2225, 2186; $\nu_{as}$(COO) 1465, $\nu_s$(COO) 1433. Diffuse reflectance spectra: $\lambda_{max}$ 270, 306 sh, 436 ($\pi$(Ru-O, Ru$_2$)→$\sigma$*(Ru-O))), 468 ($\pi$(Ru-O, Ru$_2$)→$\pi$*(Ru$_2$)), 1018 ($\delta$(Ru$_2$)→$\delta$*(Ru$_2$)), 1500 ($\pi$*(Ru$_2$)→$\delta$*(Ru$_2$)) nm.

Synthesis of [Ru$_2$(*i*-C$_3$H$_7$COO)$_4$Au(CN)$_4$]$_n$ (**3**): This compound was prepared by the reaction of [Ru$_2$(*i*-C$_3$H$_7$COO)$_4$(H$_2$O)$_2$]BF$_4$ (10.1 mg, 0.0150 mmol) and K[Au(CN)$_4$] (5.2 mg, 0.015 mmol) in a similar manner to that of **1**. Yield: 6.1 mg, 47%. Found C 28.39, H 3.02, N 6.51%. Calcd for C$_{20}$H$_{28}$AuN$_4$O$_8$Ru$_2$: C 28.21, H 3.31, N 6.58%. IR (KBr, cm$^{-1}$): $\nu$(CN) 2220, 2186; $\nu_{as}$(COO) 1471, $\nu_s$(COO) 1421. Diffuse reflectance spectra: $\lambda_{max}$ 282 br, 452 br ($\pi$(Ru-O, Ru$_2$)→$\sigma$*(Ru-O); $\pi$(Ru-O, Ru$_2$)→$\pi$*(Ru$_2$)), 1016 ($\delta$(Ru$_2$)→$\delta$*(Ru$_2$)), 1498 ($\pi$*(Ru$_2$)→$\delta$*(Ru$_2$)) nm.

Synthesis of [Ru$_2$(*t*-C$_4$H$_9$COO)$_4$Au(CN)$_4$]$_n$ (**4**): This compound was prepared by the reaction of [Ru$_2$(*t*-C$_4$H$_9$COO)$_4$(H$_2$O)$_2$]BF$_4$ (10.0 mg, 0.0138 mmol) and K[Au(CN)$_4$] (4.6 mg,

0.014 mmol) in a similar manner to that of **1**. Yield: 7.4 mg, 59%. Found C 31.45, H 3.82, N 5.98%. Calcd for $C_{24}H_{36}AuN_4O_8Ru_2$: C 31.76, H 4.00, N 6.17%. IR (KBr, cm$^{-1}$): $\nu$(CN) 2211, 2182; $\nu_{as}$(COO) 1487, $\nu_s$(COO) 1423. Diffuse reflectance spectra: $\lambda_{max}$ 278, 310 sh, 448 br ($\pi$(Ru-O, Ru$_2$)$\rightarrow\sigma$*(Ru-O); $\pi$(Ru-O, Ru$_2$)$\rightarrow\pi$*(Ru$_2$)), 1016 ($\delta$(Ru$_2$)$\rightarrow\delta$*(Ru$_2$)), 1498 ($\pi$*(Ru$_2$)$\rightarrow\delta$*(Ru$_2$)) nm.

Elemental analyses of C, H, and N were conducted with a Thermo-Finnigan FLASH EA1112 series CHNO-S analyzer (Thermo-Finnigan, Milan, Italy). IR spectra were recorded as KBr discs with a JASCO MFT-2000 FT-IR spectrometer (JASCO, Tokyo, Japan). Powder reflectance spectra were recorded with a Shimadzu Model UV-3100 UV-vis-NIR spectropho-tometer (Shimadzu, Kyoto, Japan). Magnetic susceptibility measurements were conducted using a Quantum Design SQUID susceptometer (MPMS-XL7, Quantum Design North America, San Diego, CA, USA) with a magnetic field of 0.5 T over a temperature range of 4.5–300 K. The magnetic susceptibility $\chi_M$ is the molar magnetic susceptibility per mole of [Ru$_2$(RCOO)$_4$Au(CN)$_4$] unit and was corrected for the diamagnetic contribution calculated from Pascal's constants [37]. N$_2$-adsorption measurements were conducted by a Microtrac-BEL BELSORP-mini II (MicrotracBEL, Osaka, Japan). The samples were evacuated at 298 K for 2 h prior to the measurements.

All measurements for single-crystal X-ray diffraction were made using a Bruker Smart APEX CCD diffractometer (Bruker, Billerica, MA, USA) with graphite monochromated Mo K$\alpha$ radiation ($\lambda$ = 0.71073 Å). The structures were solved using intrinsic phasing methods and refined by full-matrix least-squares methods. The hydrogen atoms were included at their positions calculated geometrically. All of the calculations were carried out using the SHELXTL software package [38]. Crystallographic data have been deposited with Cam-bridge Crystallographic Data Centre: Deposit numbers CCDC-2157503, 2157501, 2157498, and 2157500 for **1**, **2**, **3**, and **4**, respectively. Copies of the data can be obtained free of charge via http://www.ccdc.cam.ac.uk/conts/retrieving.html (accessed on 10 March 2022) (or from the Cambridge Crystallographic Data Centre, 12, Union Road, Cambridge, CB2 1EZ, UK; Fax: +44 1223 336033; e-mail: deposit@ccdc.cam.ac.uk).

## 4. Conclusions

In this study, four new heterometallic Ru$_2$Au complexes were synthesized by the reaction of tetrakis($\mu$-carboxylato)diruthenium(II,III) with tetracyanidoaurate(III). We have found that the acetate Ru$_2$Au complex and the pivalate Ru$_2$Au complex are wave-like chain molecules with the *trans*-bridging mode of the teracyanidoaurate(III) linkers, while the propionate Ru$_2$Au complex and the isobutyrate Ru$_2$Au complex are zig-zag chain molecules with the *cis*-bridging mode of the Au(CN)$_4^-$ linkers. The different bridging modes of the Au(CN)$_4^-$ linkers may come from the symmetrical difference in the substituent R groups of the carboxylato-bridges of the Ru$_2$(RCOO)$_4^+$ core. It may be considered that more symmetrical CH$_3$- and *t*-C$_4$H$_9$- groups cannot allow the steric hindrance between these alkyl groups for the *cis*-bridging mode, while the C$_2$H$_5$- and *i*-C$_3$H$_7$- groups can accommodate the *cis*-bridging mode because of the lower symmetry of the alkyl groups. The temperature dependence of the magnetic susceptibilities suggested that the magnetic interaction of the Ru$_2$(RCOO)$_4^+$ spins through the Au(CN)$_4^-$ moieties is very weak, based on the 3/2 spin state of the Ru$^{II}$-Ru$^{III}$ unit of the present complexes, which should be confirmed by the magnetization measurements at very low temperatures in further studies.

**Supplementary Materials:** The following are available online at https://www.mdpi.com/article/10.3390/magnetochemistry8050048/s1, Figure S1: Infrared spectra of [Ru$_2$(CH$_3$COO)$_4$Au(CN)$_4$]$_n$ (**1**); Figure S2: Infrared spectra of [Ru$_2$(C$_2$H$_5$COO)$_4$Au(CN)$_4$]$_n$ (**2**); Figure S3: Infrared spectra of [Ru$_2$(*i*-C$_3$H$_7$COO)$_4$Au(CN)$_4$]$_n$ (**3**); Figure S4: Infrared spectra of [Ru$_2$(*t*-C$_4$H$_9$COO)$_4$Au(CN)$_4$]$_n$ (**4**); Figure S5: Variable temperature of magnetic moment $\mu_{eff}$ for [Ru$_2$(C$_2$H$_5$COO)$_4$Au(CN)$_4$]$_n$ (**2**). The solid black line was calculated and drawn with the parameter values described in the text; Figure S6: Variable temperature of magnetic moment $\mu_{\varepsilon ff}$ for [Ru$_2$(*i*-C$_3$H$_7$COO)$_4$Au(CN)$_4$]$_n$ (**3**). The solid black line was calculated and drawn with the parameter values described in the text; Figure S7: Variable temperature of magnetic moment $\mu_{eff}$ for [Ru$_2$(*t*-C$_4$H$_9$COO)$_4$Au(CN)$_4$]$_n$ (**4**). The solid black line was calculated and drawn with the parameter values described in the text; Figure S8: Nitrogen adsorption isotherm of [Ru$_2$(C$_2$H$_5$COO)$_4$Au(CN)$_4$]$_n$ (**2**). Solid line is guide for the eye; Figure S9: Nitrogen adsorption isotherm of [Ru$_2$(*i*-C$_3$H$_7$COO)$_4$Au(CN)$_4$]$_n$ (**3**). Solid line is guide for the eye; Figure S10: Nitrogen adsorption isotherm of [Ru$_2$(*t*-C$_4$H$_9$COO)$_4$Au(CN)$_4$]$_n$ (**4**). Solid line is guide for the eye; Table S1: Some selected structural parameters of the present complexes **1–4**.

**Author Contributions:** Conceptualization, M.M.; methodology, M.M. and M.H.; in-vestigation, Y.T., D.Y. and H.T.; data curation, M.M.; writing—original draft preparation, M.M.; writing—review and editing, M.M., M.T. and M.H. All authors have read and agreed to the published version of the manuscript.

**Funding:** This research received no external funding.

**Institutional Review Board Statement:** Not applicable.

**Informed Consent Statement:** Not applicable.

**Data Availability Statement:** Not applicable.

**Conflicts of Interest:** The authors declare no conflict of interest.

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
