# Peer review of "Heterometallic Chain Compounds of Tetrakis(µ-carboxylato)diruthenium and Tetracyanidoaurate"

_magnetochemistry, doi:10.3390/magnetochemistry8050048_

Round 1
Reviewer 1 Report
The paper by Masahiro Mikuriya, Yusuke Tanaka, Daisuke Yoshioka, Motohiro Tsuboi, Hidekazu Tanaka and Makoto Handa is devoted to the synthesis, study of crystal structures and magnetic properties of [RuII(RCO2)4RuIII][Au(CN)4] complexes. It is not clear why the authors call these objects as mixed-metal complexes: it is correct to call these complexes heterometallic, since they contain atoms of two different metals (Ru and Au) in crystallographically independent positions. We can talk about the mixed-valent state of ruthenium atoms, but this is not identical to the term Mixed-metal, which is used, for example, for lanthanide complexes, for example [TbxEu1-x(L)], where the Tb3+ and Eu3+ ions occupy the same crystallographic positions and are distributed statistically.
Page 2 Scheme 1: the diagram is not neatly drawn, the dotted line between the ruthenium atoms is not parallel to the double bond. I propose to indicate the charge on ruthenium ions, and redraw the diagram in a more understandable form.
P3 Fig 1: only a small part of the spectra is informative. The enlarged fragment ~2200-2100 cm-1 should be left in the work, and the entire spectra should be placed in SI
P4F2: the transitions in the spectra must be labeled on the figure, in addition, a legend must be added to the spectrum.
P5Table 2
This table is redundant in the text and can be transferred to SI without loss.
Figures 3-10: for clarity, it would be good to place these figures side by side and partially combine, for example, molecular fragments of structures 1-4, packaging in structures 1-4.
P9L193: The values of the magnetic moments are very close. What is the measurement error? 4.24 and 4.26 coincide within the margin of error, or not?
P10Fig 11: the y-axis should be scaled from 3 to 4.5, zooming in on the physically meaningful part of the picture.
P10 Fig 12: the picture should be built in a more visual form.
Data not given in the paper (e.g. nitrogen adsorption curves for 2-4, magnetic data etc) must be reported in SI
Author Response
First, we would like to thank you for valuable suggestions for our manuscript.
As for the terminology, “mixed-metal”, we used it as the same meaning of “hetero-metal”, because it seemed to be clear the existence of the different metal ions in the complexes. But, we now understand that “mixed-metal” is not correct as the referee pointed out. We correct this terminology throughout the manuscript. Thank you for the referee for this.
Page 2 Scheme 1: According to the suggestion, we corrected Scheme 1.
Page 3, Fig.1: We added the full IR spectra in Figures. S1-S4 as supplementary data according to the suggestion.
Page 4, Fig.2: We added the transitions and legend to Figure 2. We found a minor error in the description of the transitions on the Au(CN)4 moieties during this process, and also corrected this description and reference 30.
Page 5, Table 2: According to the suggestion, we moved Table 2 to Table S1 as the supplementary data.
Figures 3-10: We combined Figures 3, 4, 5, and 6, and Figures 7, 8, 9, and 10, respectively, according to the suggestion. The revised Figures became better. Thank you for the suggestion.
Page 9, Line 193: The magnetic moments of the present complexes are very similar. The moment values coincide within the margin of error. So, we added a comment.
Page 10, Figure 11: We corrected Figure 11 according to the suggestion. It became better. Thank you for this suggestion.
Page 10, Figure 12: We corrected Figure 12 according to the suggestion.
Magnetic data and adsorption data not given in the paper were added as Figures S5—S10 in the supplementary data according to the suggestion.
The corrected part was written in red color. Revision for avoiding the duplicated overlap with previously published articles was written in brown color.
I hope that the revised manuscript will be OK.
Reviewer 2 Report
The manuscript “Mixed-Metal Chain Complexes of Mixed-Valent Dinuclear Ruthenium Carboxylate and Tetracyanidoaurate” by Masahiro Mikuriya, Yusuke Tanaka, Daisuke Yoshioka, Motohiro Tsuboi, Hidekazu Tanaka, Makoto Handa is devoted to the synthesis and characterization of the four mixed-metal complexes of tetrakis(µ-carboxylato)diruthenium(II,III) with tetracyanidoaurate(III) [Ru2(RCOO)4Au(CN)4]n, where R = Me, Et, i-Pr, and t-Bu.
The content of paper expands the range of heterometallic compounds containing mixed-valent diruthenium synthon. The manuscript is written in good language, which is essential for a good understanding of the material.
The strongest side of the article is that the authors managed to obtain single crystals of all compounds and determine their crystal structure, which confirmed the results of their IR spectroscopic study.
However, there are both significant and minor flaws in the design and content of the article.
Design notes:
The figures illustrating structure of the complexes are made in different styles, especially the color of metal ions for packing diagrams and ORTEP views. For the latter, it is necessary to present an independent structural unit.
In Table 2, the values of the structural parameters are given close to the designations of these parameters, which greatly complicates the perception of the information contained in the table.
The authors present "live" experimental data for variable-temperature magnetic moment and adsorption isotherms only for the compound 1, stating that for the other compounds the corresponding curves are similar. However, this is not enough, and the experimental curves must be submitted as a supporting information.
Content notes:
Line 43 - replace dicyanoargentate with dicyanidoargentate.
Line 54 - replace "a two-dimensional sheet" with "a layer".
Line 66 - fix Ruthenium(II,IIII).
Line 112. The authors state "Crystals suitable for X-ray diffraction work were grown by slow diffusion of water into aqueous solutions of reaction materials". This is very strange, since the synthesis is carried out in an aqueous solution. Maybe it was not diffusion that was meant, but the slow evaporation of the solvent.
Line 242. Estimate "n" and give the theoretical CHN values for it.
Very huge doubts are caused by the part devoted to magnetic properties. The authors give values of the magnetic moment only at room temperature, arguing that values in the range 4.24-4.29 μB correspond to spin 3/2, although the theoretical value for g=2 and S=3/2 is 3.873, which is much lower. In addition, the introduction lacks information on the magnetic properties of previously studied related systems.
The authors model the magnetic behavior of their objects only based on the temperature dependence, which cannot provide a real picture of interactions. This picture can only be provided by simultaneously simulating both the temperature dependence of the susceptibility (magnetic moment) and the magnetization curve at low temperature (2 K). In addition, the latter gives unambiguous data on the ground spin state of the paramagnetic system.
Since I was unable to obtain crystallographic data from the CCDC for the studied compounds, and the authors did not present them as support information, I cannot draw any conclusions about the quality of the structural information.
Considering the above remarks, and the fact that the paper is supposed to be published in "Magnetochemistry" without the presentation of low-temperature magnetization curves for heterometallic diruthenium complexes, this manuscript cannot be accepted for publication.

Author Response
Author’s Reply to the Review Report (Reviewer 2)
First, we would like to thank you for valuable suggestions for our manuscript.
According to the figures illustrating structure of the complexes, we tried to show only the asymmetric unit for the ORTEP views according to the suggestion. However, these figures are very difficult to see the molecular structures for the readers. Therefore we showed the molecular fragments for the ORTEP view showing the asymmetric unit without the symmetry codes and the other part with the symmetry codes. Moreover, the symmetry codes were changed to small roman numbers to be recognized easily. The color of the metal ions was also changed to be in the same style.
Table 2 is complicated as pointed out. Therefore, we corrected Table 2 clearly and was moved to Table S1 as the supplementary data.
According to the suggestions, the variable-temperature magnetic moments and adsorption isotherms of the compounds 2, 3, and 4 were given in Figures S5-S7 and S8-S10, respectively.
Line 43, Line 54, Line 66, Line 112: We corrected according to the suggestions.
Line 242. We have already estimated n = 1 and gave the theoretical values. We corrected the description “1·nH2O”, which was confusing for the readers.
As for magnetic properties, we added a comment on the magnetic properties of dinuclear ruthenium(II,III) carboxylates and their derivatives in the introduction. These complexes show the room-temperature magnetic moments in the range of 3.6—4.4 µB per RuII-RuIIIunit, a little higher than the spin-only value for S = 3/2 spins. These values have been considered as the S = 3/2 ground state by many researchers and confirmed the state by EPR measurement by some researchers, and thus most of the papers dealing with the magnetic properties of these complexes have been reported based on the magnetic susceptibility measurements without the magnetization measurements. Although I can understand the importance of the magnetization data, we could not have enough liquid helium to measure the magnetization data unfortunately. We added a comment on the magnetization data in future work in conclusion.
The corrected part was written in red color. Revision for avoiding the duplicated overlap with previously published articles was written in brown color.
I hope that the revised manuscript will be OK.
Reviewer 3 Report
The manuscript by M. Mikuriya et al. describes the synthesis and characterization of four new polymeric compounds composed of dinuclear ruthenium carboxylates connected by [Au(CN)4]- anions. The new compounds have been characterized by elemental analysis, IR and UV-vis spectroscopies and single crystal X-ray structure determination. Their magnetic susceptibility measurements are also reported.
The synthetic procedures are straightforward and adequately described. The X-ray structural characterization is of high quality. The rest of the experimental data are routine and unsurprising: the magnetic properties are consistent with those reported earlier for similar compounds, and the lack of porosity is also expected by the rather tight packing of the polymeric structures. This is a rather routine piece of work, but correct nevertheless, therefore publishable.
My only question is on the assignment of the C-N stretches in the IR spectra: how do the authors know that the higher energy bands are attributed to the bridging CN groups? One might have expected that the increased back-donation into the antibonding CN-orbitals of the bridging cyanides would have caused their stretching mode to be a lower energy than those of the terminal cyanides.
Author Response
Author’s Reply to the Review Report (Reviewer 3)
First, we would like to thank you for valuable suggestions for our manuscript.
As for the assignment of the CN stretching band, we added a comment on the small contribution of the back-donation in the CN compounds.
The corrected part was written in red color. Revision for avoiding the duplicated overlap with previously published articles was written in brown color.
I hope that the revised manuscript will be OK.
Reviewer 4 Report
This paper describes mixed-metal chain complexes of mixed-valent dinuclear Ruthenium Carboxylate and Tetracyanidoaurate
The main content of the paper, describes four crystal structures of such complexes. The refinements are all well done and of a good standard.
However, the description of the structures could be improved and the labelling of atoms made more consistent. For example in 1, the oxygens in the two ligands are labelled O1, O4 and O2, O3 while in the similar structure 4, they are labelled O1, O2 and O3, O4 which is a more sensible system which should be followed throughout. It would be sensible to renumber the structures so that 4 is called 2 and vice versa.
In regard of the statement about voids, (line 141) the authors find voids in structure 1 but the checkcif file finds voids in structure 4 but not in structure 1. The authors also say that structure 4 does not have voids!! (line 159). The authors need to explain this disparity. It seems to me that it is wrong to draw conclusions about cavities from 2D diagrams.
Surely it is necessary to have the figures for the structures in the same style. Thus while figs 1 and 2 are equivalent, figs 7 and 8 are different from these two, the former having thinner lines for bonds and the latter having thicker lines.
Figure 4 is a pretty figure but how much chemical interest is there? What does it show? The colour scheme doesn’t help nor does the lack of labelling
Figure 6 is poor. Atoms cut in half are unacceptable. The authors need to redo the figure including complete ligands. Similarly Figures 9 and 10 are poor with the edges of the diagram showing incomplete molecules.
It is noticeable that in this section there is no comparison with previous crystal structures. How unique are the present structures?
The authors describe the structures as containing Ruthenium(II,III) which implies that the two Ru atoms are different but surely it should be pointed out that the electronic configuration is more complicated than that, and in all structures the two Ru atoms making up the cluster are equivalent. Indeed in most structures they are crystallographically equivalent.
Surely 15 self-citation references is far too many and needs to be reduced.
The paper contain interesting work but the presentation should be improved before publication
Author Response
Author’s Reply to the Review Report (Reviewer 4)
First, we would like to thank you for valuable suggestions for our manuscript.
As for the inconsistent numbering of the carboxylate-oxygen atoms, this came from the different symmetries of the complexes 1—4. So, we would like to keep this numbering. We corrected the description of the crystal structures in clear.
As for the statement about voids, we described it erroneously from our misunderstanding. We correct the description according to the suggestion. We thank for the valuable suggestion.
As for the structures in the same style, we corrected Figures according to the suggestion.
As for Figures 4, 6, 9, and 10, we agree with the suggestion, and we corrected these according to the suggestions.
As for the structural comparison with the previous crystal structures, we added a comment according to the suggestion.
As for the equivalent mixed-valent state of the RuIIRuIII core, we noted this at the end of the description of the crystal structures according to the suggestion.
According to the suggestion, we reduced the self-citation references by deleting five references.
The corrected part was written in red color. Revision for avoiding the duplicated overlap with previously published articles was written in brown color.
I hope that the revised manuscript will be OK.
Round 2
Reviewer 1 Report
The authors took into account the comments and an article can be published.
Reviewer 2 Report
The authors took into account all the comments, significantly improved both the text of the article and the figures. Supporting Information has also changed for the better.
In this form, the manuscript can be accepted for publication.
